# User preferences for an mHealth app to support HIV testing and pre-exposure prophylaxis uptake among men who have sex with men in Malaysia

Lindsay Palmer[1], Jeffrey A. Wickersham[2], Kamal Gautam [3], Francesca Maviglia[2], Beverly-Danielle Bruno[4], Iskandar Azwa[5,6], Antoine Khati[3], Frederick L. Altice[2], Kiran Paudel [3], Sherry Pagoto[3], Roman Shrestha [2,3,5]*

1 Department of Population and Quantitative Health Sciences, UMass Chan Medical School, Boston, Massachusetts, United States of America, 2 Department of Internal Medicine, Section of Infectious Diseases, Yale School of Medicine, New Haven, Connecticut, United States of America, 3 Department of Allied Health Sciences, University of Connecticut, Storrs, Connecticut, United States of America, 4 Rollins School of Public Health, Emory University, Atlanta, Georgia, United States of America, 5 Centre of Excellence for Research in AIDS (CERiA), Faculty of Medicine, University of Malaya, Kuala Lumpur, Malaysia, 6 Infectious Diseases Unit, Department of Medicine, Faculty of Medicine, University of Malaya, Kuala Lumpur, Malaysia

* roman.shrestha@yale.edu

**Data Availability Statement:** All excerpts of the transcripts relevant to the study are within the manuscript.

## Abstract

Recent estimates report a high incidence and prevalence of HIV among men who have sex with men (MSM) in Malaysia. Mobile apps are a promising and cost-effective intervention modality to reach stigmatized and hard-to-reach populations to link them to HIV prevention services (e.g., HIV testing, pre-exposure prophylaxis, PrEP). This study assessed attitudes and preferences toward the format, content, and features of a mobile app designed to increase HIV testing and PrEP uptake among Malaysian MSM. We conducted six online focus groups between August and September 2021 with 20 MSM and 16 stakeholders (e.g., doctors, nurses, pharmacists, and NGO staff) to query. Transcripts were analyzed in Dedoose software to identify thematic content. Key themes in terms of app functions related to stylistic preferences (e.g., design, user interface), engagement strategies (e.g., reward systems, reminders), recommendations for new functions (e.g., enhanced communication options via chat, discussion forum), cost of services (e.g., PrEP), and legal considerations concerning certain features (e.g., telehealth, patient identification), minimizing privacy and confidentiality risks. Our data suggest that a tailored HIV prevention app would be acceptable among MSM in Malaysia. The findings further provide detailed recommendations for successfully developing a mobile app to improve access to HIV prevention services (e.g., HIV testing, PrEP) for optimal use among MSM in Malaysia.

## Author summary

In our study, we aimed to develop a mobile health app to support HIV testing and PrEP use among MSM in Malaysia. Given the significant social stigma and legal challenges

**Funding:** This work was supported by the National Institute on Drug Abuse (K01 DA051346 to RS) and the Fogarty International Center (R21TW011665 to RS). The funders had no role in study design, data collection and analysis, decision to publish, or preparation of the manuscript.

**Competing interests:** The authors have declared that no competing interests exist.

faced by MSM in accessing healthcare, we wanted to understand what features and functions would make such an app most useful. We gathered insights from MSM and community health workers, including doctors and NGO staff, through online focus groups to understand their preferences for the app's features and functions. Participants desired functions such as ordering HIV self-testing kits, connecting with healthcare providers, scheduling online appointments, viewing test results, doing e-consultations, ordering PrEP medications, and tracking daily PrEP intake. They also valued informational resources, healthcare provider locators, and communication features like live chat and discussion forums. Engagement features like notifications, reminders, and customizable content were also important. However, there were concerns about ensuring the app's cultural relevance, maintaining user-friendliness despite numerous features, protecting user privacy, and making the app visually appealing and discreet to avoid disclosing the user's sexual orientation. We found that a well-designed, culturally sensitive app with these features could significantly improve access to HIV prevention services for MSM in Malaysia.

## Introduction

The HIV epidemic in Malaysia disproportionately affects men who have sex with men (MSM) [1]. Currently, in Malaysia, 21.6% of MSM live with HIV, and prevalence is on the rise [2]. Same-sex sexual behavior is criminalized in Malaysia by both secular and religious law [3], which has severe downstream consequences for the health of MSM that require urgent redress [4].

MSM face a complex combination of individual, interpersonal, and societal barriers to accessing HIV prevention services, which are typically delivered in person. Many Malaysian MSM do not have a comprehensive awareness of HIV transmission and prevention [5,6]. In addition, Malaysian MSM may internalize the stigma of having HIV (e.g., fear of becoming HIV positive) and being a sexual minority [5,6]. Barriers to comprehensive HIV care for MSM include widespread sexual prejudice and HIV stigma that manifests in discrimination within the healthcare system [7–10]. Compounding these personal and social barriers, MSM in Malaysia also face systemic and policy-level barriers related to the criminalization of same-sex sexual behavior (e.g., sexual activity) [3,11]. Technology-based interventions are a promising and innovative avenue to circumvent and navigate the barriers to in-person HIV prevention services among MSM.

Technology-based interventions, particularly those leveraging mobile apps, have shown promise as innovative, cost-effective approaches to improve health outcomes across various conditions and settings [12–14]. A systematic review of mHealth interventions for HIV prevention among MSM, identified 16 published studies, and 11 of those studies were conducted in the U.S., which highlights the need to generate knowledge on developing interventions outside of a U.S. context [15]. Few randomized controlled trials of mobile apps geared toward increasing knowledge on safer sexual behavior and regular HIV self-testing among MSM, participants receiving the app had a reduction in the number of condomless anal sex and higher rates of HIV self-testing compared to a control group of participants [16–18]. In the context of HIV prevention in Malaysia, mobile technology offers an opportunity to reach stigmatized and hard-to-reach populations, like MSM, and link them to HIV prevention services (e.g., HIV testing, pre-exposure prophylaxis, PrEP) [19,20]. Given that 78.0% of Malaysians reported smartphone use in 2018 and 94.6% of smartphone users reported accessing the internet through their smartphones, it is reasonable that a mobile app-delivered intervention is

feasible in a Malaysian context [21]. MSM in Malaysia have nearly universal smartphone access and a high willingness to receive mobile app-based interventions [20,22]. Notably, a systematic review and meta-analysis of 16 studies demonstrated the promise of mobile app-based HIV prevention services for MSM in improving health outcomes and their acceptability among MSM [15]. Mobile app interventions may appeal to MSM in Malaysia because they allow the intervention to be delivered privately, confidentially, and conveniently, which may abate stigma-related concerns.

The current qualitative study examines user preferences in an app designed to support users in preventing and treating HIV among MSM in Malaysia, which addresses a gap in the literature, given that most research about mobile app-based interventions has been examined in high-income countries. Strategies developed for populations within high-income countries may not be suitable for low and middle-income countries (LMIC). Research to improve the health of stigmatized and vulnerable groups within resource-limited settings is urgently needed [23]. As a first step towards creating a mobile app tailored for HIV prevention in LMIC settings, we examined both MSM and stakeholders' perspectives regarding preferred features and functions that will maximize the uptake of HIV prevention services and user engagement and satisfaction. Findings from the current study can inform future digital intervention content, especially for vulnerable populations within other LMIC settings.

## Methods

### Participants and settings

Between March and May 2021, we recruited 36 participants (20 MSM and 16 stakeholders) for focus groups (FG). MSM in Malaysia were recruited through LGBT-friendly non-government organizations (NGOs) using flyers and emails and through advertisements on social networking apps (e.g., Grindr, Hornet). Eligibility criteria included: a) being 18 years or older; b) being self-identified as MSM; c) being able to understand English or Bahasa Malaysia and d) currently residing in Malaysia. Inclusion criteria for stakeholders were: 1) age 18 years or older, 2) currently employed as physicians, nurses, pharmacists, mental health counselors, community outreach workers, and NGO staff involved in providing HIV-related services to MSM. Stakeholders were recruited through local NGOs and clinics/hospitals.

### Procedures

We conducted six virtual FG sessions (three among MSM and three among stakeholders) using a video-conferencing platform. Each session lasted about 90 minutes and included an average of 6 participants ranging from 4–9 participants per session. Each session was audio-recorded and transcribed. A trained facilitator led the session, while a co-facilitator took notes, recorded non-verbal cues, and typed responses on a virtual whiteboard. Each participant provided verbal consent before starting the session and was compensated for their participation (RM 40 for MSM and RM 80 for stakeholders; US $1 ~ RM 4). We took several steps to protect the privacy of our participants. First, each participant was provided with their own unique invitation to a password-protected online session. Second, before entering the online FG session, each attendee was placed in a waiting room where their participation was verified before entering the FG. In the waiting room, researchers also instructed participants in additional privacy measures such as changing their display names to pseudonyms and turning off their cameras to ensure privacy and confidentiality. The study protocol was approved by the Institutional Review Boards at the University of Connecticut and the University of Malaya Research Ethics Committee.

Participants also completed a brief demographic and behavioral questionnaire via Qualtrics before the FG session. The primary purpose of the FG session was to gain insight into participants' perspectives on a prospective mobile app designed to deliver HIV-related services to inform the development of this mobile app. During the discussion, a semi-structured topic guide was used to ask participants about attitudes toward the app's format, content, and features to improve access to HIV prevention services among Malaysian MSM. Specifically, in the first part of the discussion, participants were questioned about the facilitators and barriers of HIV testing and starting PrEP for MSM. In the last part of the discussion, participants were shown the prospective app "HealthMindr" and given a demonstration of its current functions and features. Following the demonstration, participants were asked to provide feedback on the features, content, and format of HealthMindr, focusing on its adaptation for MSM in Malaysia [24]. Questions were consistent across FG with slight differences in wording as needed to be relevant to the population in the FG. Feedback was elicited on their thoughts on the appearance and functionality of the app interface, appeal, usability, components they would like and dislike, and components to increase engagement and user retention.

## Data analysis

SPSS Version 29.0.0 software was used to calculate descriptive statistics (frequencies and percentages) for the survey data. Transcriptions of the FG sessions were coded by two coders (FM, BB) using the FG session and entire discussion as the unit of analysis [25,26]. First, the two coders read and re-read transcriptions for each question to identify key recurring patterns. These initial codes were then used to inform a coding scheme for each question that was then applied to the transcript. Then two coders coded responses independently to minimize any coding biases. Two debriefing meetings between coders were held so that coders could resolve discrepancies and reach consensus. After the coding meetings, a final coder (LP) finalized categories from the codes into parent codes representing larger categories and child codes within each larger subset using an arbitration process [27]. To establish themes, codes were collated to establish important overarching categories that represented patterns in the dataset [28]. Themes were then validated through discussion until consensus was reached. Each theme is presented in the results sections with example quotes to best illustrate the findings.

## Results

Among MSM, half (50%, n = 10) were Chinese, 40% (n = 8) were Malay, one was Indian (5%), and one was mixed ethnicity (5%). The majority (75%, n = 15) reported being sexually active in the last 6 months, although only 10% (n = 2) reported using condoms all the time. Most participants reported having previously tested for HIV, with 85% (n = 17) having tested in the last six months. The majority of the MSM sample (n = 16, 80%) had heard of PrEP, 12 (60%) had taken PrEP, and 8 (40%) were currently on PrEP (Table 1).

Of the 16 stakeholders, half (50%, n = 8) were Chinese. Most of them were physicians (38%, n = 6), followed by registered nurses (19%, n = 3) and healthcare administrators (19%, n = 3). These stakeholders worked at community-based organizations (33%, n = 5), private clinics (33%, n = 5), academic or university clinics (19%, n = 3), and government clinics (13%, n = 2).

The FG session involved three parent themes composed of child codes (Table 2) that address: 1) app features and functionalities; 2) additional attributes that might increase user engagement and retention; and 3) areas that participants identified as important considerations when developing an app to serve Malaysian MSM.

**Table 1. Participant characteristics.**

| MSM (n = 20) | | Stakeholders (n = 16)* | |
|---|---|---|---|
| **Variable** | **n (%)** | **Variable** | **n (%)** |
| Ethnicity | | Ethnicity | |
| *Chinese* | 10 (50%) | *Chinese* | 8 (50%) |
| *Malay* | 8 (40%) | *Malay* | 3 (19%) |
| *Indian* | 1 (5%) | *Indian* | 3 (19%) |
| *Mixed* | 1 (5%) | *Native Sabahan* | 1 (6%) |
| Sexual Orientation | | Occupation | |
| *Gay/PLU* | 20 (100%) | *Outreach Worker* | 1 (6%) |
| Sexual Activity (Past 6 months) | | *Administrator* | 3 (19%) |
| *Yes* | 15 (75%) | *Medical Doctor* | 6 (38%) |
| *No* | 5 (25%) | *Registered Nurse* | 3 (19%) |
| Condom Use (Past 6 months) | | *Pharmacist* | 1 (6%) |
| *Never* | 2 (10%) | *Psychologist* | 1 (6%) |
| *Sometimes* | 2 (10%) | Type of Health Facility | |
| *Most of the time* | 9 (45%) | *Community-based Organization* | 5 (33%) |
| *All the time* | 2 (10%) | *Clinic/Hospital: Private* | 5 (33%) |
| *No Response* | 5 (25%) | *Clinic/Hospital: Academic* | 3 (19%) |
| Substance Use (Past 6 months) | | *Clinic/Hospital: Government* | 2 (13%) |
| *Alcohol* | 5 (25%) | Facility offers HIV Testing | |
| *Cigarettes* | 2 (10%) | *Yes* | 14 (88%) |
| *Crystal Meth* | 3 (15%) | *No* | 1 (6%) |
| *GHB/GBL* | 1 (5%) | Involved in providing PrEP services | |
| *Poppers* | 2 (10%) | *Yes* | 11 (69%) |
| *Marijuana* | 1 (5%) | *No* | 4 (25%) |
| *None* | 10 (50%) | Number of MSM patients on PrEP | |
| Tested for HIV | | *0–10* | 4 (25%) |
| *>6 months ago* | 2 (10%) | *11–50* | 9 (56%) |
| *≤6 months ago* | 17 (45%) | *51–100* | 0 (0%) |
| *Never* | 1 (5%) | *100* | 2 (13%) |
| HIV Status | | | |
| *Negative* | 19 (95%) | | |
| *Unknown* | 1 (5%) | | |
| Ever heard of PrEP | | | |
| *Yes* | 16 (80%) | | |
| *No* | 4 (20%) | | |
| Ever taken PrEP | | | |
| *Yes* | 12 (60%) | | |
| *No* | 8 (40%) | | |
| Currently on PrEP | | | |
| *Yes* | 8 (40%) | | |
| *No* | 12 (60%) | | |

*Out of the 16 participants, 1 did not complete the Qualtrics survey

**Table 2. Codes and definitions.**

| Parent Codes | Child Codes | Description | MSM | Stakeholders |
|---|---|---|---|---|
| Preferences and Desires for the Apps Functions | Order management | Online order of healthcare products, such as medication (i.e., PrEP), HIV self-testing kits, harm reduction products (e.g., condoms, lubes), delivery and pickup options; order tracking | ✓ | ✓ |
| | e-Visits | Connect with health care providers, including mental health professionals, and get medical advice within the app | ✓ | ✓ |
| | Test results | The ability for users to view test results; receive timely notification of results for tests | ✓ | ✓ |
| | Medication tracker | Tracking of and reminders of medications (i.e., PrEP) or any other medications the user may be taking. This includes visual trackers, personalized medication reminders, prescription refill reminders, medication progress report | ✓ | ✓ |
| | Informational Multimedia Resources | Multimedia and resources on HIV self-test kits, substance use, HIV testing, PrEP, "window period," and other HIV resources. In addition, this feature would provide LGBTQ+ news, HIV vaccine progress & other LGBTQ+-relevant health research written in layperson's terms. | ✓ | ✓ |
| | Healthcare provider locator | A feature that catalogs nearby clinics with their contact information (e.g., location, hours), availability (e.g., appointment openings, services provided), the relative cost of services, and whether the location is "LGBTQ+ friendly." | ✓ | ✓ |
| | Additional support services | Additional features would include a personal risk behavior assessment to advise users on testing frequency, contact tracing for potential HIV exposure, mental health services (e.g., crisis services, mood tracking) | ✓ | ✓ |
| | Communication | Live chat with healthcare to allow two-way communication with providers, text messaging with providers, and Peer support/discussion forum to be able to discuss stigmatized and/or health-related topics | ✓ | ✓ |
| Preferences and Desires for User Engagement | Notifications / reminders | Ability to set up notifications/reminders for appointments, testing, and other facets of care. | ✓ | ✓ |
| | Interactivity | Elements of gamification (e.g., rewards, incentives) | | ✓ |
| | Customization/personalization | Adding search features and allowing customizable content (e.g., background colors). | ✓ | |
| Caveats and Concerns | Culturally relevant | The app must be sensitive to Malaysian cultural norms | ✓ | ✓ |
| | Too many features may jeopardize user-friendliness | While an expansive range of features and functions may be desirable, it can also make the app's interface difficult to use. | | ✓ |
| | Privacy / confidentiality | Data must be secured to protect the confidentiality of the users | ✓ | ✓ |
| | Visually appealing and discreet | The app logo and/or features cannot convey the sexual orientation of the user, nor can the app be construed as an app for MSM | ✓ | ✓ |
| | Language | Ability to change language settings | ✓ | ✓ |

## Preferences and desires for the apps functions

### Order management

MSM participants reported that the app would be an excellent modality for purchasing safe sex supplies, self-testing kits, and other sexual health products conveniently and confidentially. However, participants did raise the concern that discreetly delivering the products could be challenging. In particular, MSM participants discussed methods of delivering the products anonymously. For example, one participant suggested using a common delivery station rather than personal home addresses:

> *For the delivery of [HIV] test kits, apart from just having my personal address, maybe there is a collection site and the way I can actually go and collect rather than me—rather than it being delivered to my house because, like for some who are staying with their family, they may rather obtain it somewhere else.*

> *[MSM, FG2)*

Stakeholder participants also indicated the need for a delivery or order management feature within the app. They suggested this might make it easier for the users to receive medication and safer sex supplies. One participant pointed out that their clinic has been developing a similar service, and they thought that the app could help facilitate scaling up of such service:

"*Yeah we have been experimenting a little bit with that in our clinic as well, maybe the app can help streamline the process and make it a little bit more organized.*"

*[Stakeholder, FG 1]*

Participants supported features that would manage or even permit ordering products related to HIV testing, prevention, and treatment via the app.

### e-Visits

Participants stated they would like the app to facilitate healthcare services (e.g., video consultation). Many MSM participants acknowledged that a significant benefit of app integration is convenience. Utilizing e-Visits can reduce clinic waiting times and expedite connections between institutions that require communication to coordinate patient care.

*Consultation as guidance in the beginning, yeah, so when you say, do you have the video consultation, then everybody who wants to do it [video consultation] must be aware. . . this feature using the app is very good for discreet people. . . who refuse to go to the clinic because they have to wait or probably simply can't go to the clinic because it was during office hours.*

*[MSM, FG 3]*

Stakeholders shared similar sentiments, suggesting that the app can conveniently deliver services (e.g., prescription renewal). One stakeholder suggested that the app could be comprehensive in terms of supporting the wellness of users by offering them a variety of services:

*But it will be a very convenient app where they could, you know, talk to their doctor, get their medications, know about testing, and then have video consultations. Record their sexual exposures, you know, where it is all their where they get information, so they do not have to go to various places or various sites to look for this information and it's all there in one particular app. So, I think it will be an added value in our PrEP clients' daily lives.*

*[Stakeholder, FG 1]*

However, physician stakeholders primarily recognized the importance of addressing potential legal issues in integrating a telehealth component into the app. One stakeholder referenced the Telemedicine Act of 1997 as Malaysian legislation that must act as a guideline during the app development and implementation and inform app users of the consent policy to ensure understanding between patient and provider on the expectations for general information and result dissemination. Health institute liability was also a concern in cases of misguided consultation provided by medical practitioners, with stakeholders expressing the need for lawyers' involvement to ensure personnel coverage. Moreover, physicians were skeptical about the confidentiality and authentication measures upheld by the app, particularly concerning the maintenance of personal health information (PHI):

*Identification, I am starting to feel like it is going to be a little bit of a problem, right? So, I have seen this patient, I have drawn his blood, he logs on, his video is not on.*

[Stakeholder, FG 1]

*I do not know whether I'm speaking to the right person, you know. So maybe that needs to be a way to either force the video to be on or encourage the video to be on so I know I am talking to the right person or some way to verify that this is the correct person, you know. What if an email lands in the wrong hands and yeah, so I think those basic things are quite important.*

[Stakeholder, FG 1]

### Test results

Both MSM and community stakeholder participants reported that it would be suitable to provide users with the ability to view laboratory test results in a timely manner. Participants pointed out that this would greatly expedite care, meaning that participants could receive treatment or take precautions more quickly. One participant pointed out,

"*I would love to get my results through the apps, I mean it's easier for me to track back, whatever it is.*"

[MSM, FG 2]

However, some stakeholders felt uncomfortable with the legal implications of providing test results online via the app. Stakeholders were concerned about whether patients could perform and obtain accurate results from some of the tasks required to complete self-testing measures. Others were more open to accompanying video consultations if the tests were conducted again after initial use, depending on when the test was taken concerning the window period of potential exposure:

"*The kind of patient that we always see that comes into the clinic to do a rapid test are patients who fall within the window period. Right, so if they fall within the window period, there is aneed for them to repeat the test again. Right. So, if let's say you are not linking them to the clinicians, there is a high chance that you might misdiagnose them or give false negative results.*"

[Stakeholder, FG 1]

### Medication tracker

Participants reported that using an app for medication management would greatly benefit those who take daily medication. A simple component that participants discussed was that the app could send them push notifications to remind them to take their medication. In addition to simple medication reminders, participants reported that the app could make taking daily medications easier by offering a variety of tools including reminders to schedule follow-up appointments and renew prescriptions. Stakeholders felt that the app could benefit from a "medication diary" that can help participants take their medication more consistently. One stakeholder pointed out,

"*Do I mean—do [sic] the app have any diary like medication diary to remind them that they have taken the medications and stuff like that? If let's say patients—I mean, clients—are on a*

*daily basis PrEP. And that would be very, very helpful because most of the time, many patients are actually on their handphones, and it could actually pop up suddenly, and they would, you know, it reminds them that they have to pop their PrEP in."*

*[Stakeholder, FG 1]*

### Informational multimedia resources

Participants stated the desire for the app to educate users, especially in light of limited, widely available information concerning PrEP, testing procedures, treatment options, and points of access in Malaysia. Participants reported that it would be advantageous for the app to provide educational content on various health concerns, including other health topics that concern LGBTQ+ people in Malaysia (e.g., substance use). One participant discussed that the app could be a source of health information that educates users. However, it also could be a reliable source to combat misinformation and possibly stigma related to the use of PrEP:

"*Not only, particularly in LGBT community news, but also like, you know the progress of maybe there's an HIV vaccine what is the progress and everything like that because, generally, I think. Like you, you always YouTube all these things, but you don't know whether it's real or not, you know everyone will have their own speculation, everything like that, but if this App could, particularly you know, give us a lot education wise and everything like that so at least we can share to friends that we can start to you know tell other people, because sometimes I do have that problem itself like if I take PrEP people will say oh you have HIV you have HIV or something like that. So, I think if you could provide more education wise information.*"

[MSM, FG 2]

In terms of PrEP, stakeholders recognized the importance of an easily comprehensible educational component focused on PrEP that includes discussion of potentially interconnected matters such as substance use. Specifically, the presentation of information should be in plain language to ensure that the user can understand and be able to make the best of the app.

"*I think it [mobile app] should have resources that are easy to understand and address substance use. It is highly prevalent in the population [MSM] and we can never talk about PrEP without discussing substance use.*"

[Stakeholder, FG2]

### Healthcare provider locator

Participants noted that a valuable component of the app would be to help users manage their care. In particular, both stakeholders and MSM reported that they felt users would benefit from features that detail the pricing of services at different clinics, whether the clinics were LGBTQ+ friendly, as well as the operating hours of nearby clinics/hospitals. One MSM participant detailed that it would help participants prepare for their appointments:

"*So, some of the options, I think most of the options that I saw were something that I would like to look forward to into the app itself. So, something like a way I can get HIV testing, where I can get prep from then also additionally, like which clinics are LGBT friendly will be*

*nice as well, especially in Malaysia. Then yeah probably if I think I didn't see this, but I think if it's possible, probably getting the pricing on the testing and the PrEP medication in certain clinics would be great as well, and these people can prepare beforehand about how much they are about to spend there."*

*[MSM, FG 2]*

## Additional support services

In addition to providing sexual health services, participants pointed out the need for other support services. For example, one MSM participant pointed out that the app can serve to support various health needs,

*". . .drug use, mental health, basic human rights for gays in Malaysia? Yup. basic screening mental health. for drug use maybe umm. . .way to reduce harm while taking substances."*

[MSM, FG 1]

A theme among MSM was that the app provides mental health resources, such as telehealth services to be connected with a provider. MSM supported these app features and suggested areas for expanding these types of services offered. One MSM participant articulated that the app could help address the need for mental health services among the LGBTQ+ community in Malaysia:

*"Generally, I agree with what [redacted] said, but I think what really caught my attention was the availability for the counseling services, I think that is really, really good actually, but I think. The LGBTQI community in Malaysia I think they don't really have a channel to seek counseling, or it's not easily accessible for them or something. And something they should talk about something they would need help in terms of mental health, but I don't think it's available so having this on the app is actually very helpful."*

*[MSM, FG 2]*

## Communication

Participants emphasized the need for communication features, where app users could interact with fellow peer app users and healthcare providers. Potential MSM users found the idea of discussion forums helpful in providing education on HIV-related information in a judgment-free environment with others who can understand each other's experiences. A few participants found that it may not be necessary, as the chat would likely be monitored by professionals, making users feel less inclined to be fully authentic.

Stakeholders promoted having chat rooms to minimize redundancy in answering questions that patients may have already asked earlier.

*"A lot of times, it becomes very repetitive, when we are answering the same question [and] giving out the same information, but to many different clients. But if you have sort of like a chat room, then any new users can just scroll up back to the top or have a list of questions that been posted [and] they can click on it, and then the answer would already be there, so that sort of minimizes time that clinic staff spend answering questions over the phone, or the WhatsApp*

*or email or Instagram—we have all these platforms, and the same questions keep getting asked again and again. So, chat room would be amazing where they can, sort of like, [have] the FAQ section that the previous APP that we saw had—something like that—but you know with a bit more life with clients [that] can send in questions and then we post answers."*

[Stakeholder, FG 3]

As this provider observed, patients often have similar questions, and a chat feature could maximize appointments and save providers time because it enables patients to receive answers to common questions quickly and easily. A record of such interactions also helps fellow patients who may share the same questions but are more hesitant to vocalize them. Physicians also find the chat function helpful in facilitating direct messaging with patients, reducing the burden on clinics to allot set times for patients to come in with questions and allowing a better consultation flow. However, expected response times would need to be established within the app to ensure an appropriate work balance for medical professionals. This group has expressed concerns over expectations of 24/7 availability to patients.

## Preferences and desires for the app interface

### Notifications and reminders

While a desire for notifications and reminders were important to participants within other categorical content, this category also stood out as a tool for user engagement and retention. In general, we found support for features such as notifications and reminders that would engage the user in the app. As one participant pointed out,

"*It would be good if there is a section for notifications and reminders, for example if we have a doctor appointment or when we take the PrEP medication the apps will inform us whether we already have taken the medication or not lie my activities, it informs us on our current status.*"

[MSM, FG3]

Participants discussed whether this "notification and reminder section" could be a stand-alone feature that is customizable based on the users' needs. Stakeholders cautioned that this feature would also need to be customized based on the user's needs. One stakeholder said,

"*I think the most important thing about that is not everyone wants their notifications publicized and you know, made known on the screen of your smartphone.*"

[Stakeholder, FG3]

### Interactivity

Stakeholders have mentioned the value of including a loyalty reward system or using gamification to ensure user engagement and retention. For example, incentivizing or rewarding (e.g., points and badges) users for completing certain tasks on the app, such as consistently tracking medication intake and screening for mental health illness. Users can redeem points for cash, HIV test kits, or PrEP medications, and unlock badges such as avatars by completing specific tasks. One stakeholder mentioned that this was a system that seemed very effective in another app that they were aware of:

*"Personally, I have gone through another app made by REDACTED from the REDACTED called REDACTED which stands for REDACTED. Yeah, I'm sorry if you may be a bit bias, but the thing I like about that app is it has a sort of hamster wheel that keeps patients going which encourages you to check in that you've taken your medication today. And it rewards you for that milestone, so it encourages you to update the app, you have taken a medication, which encourages compliance and adherence."*

[Stakeholder, FG 3]

## Customization/Personalization

MSM participants reported that they thought the app would benefit from features that make the app customizable and feel more personal. For example, search features or other options to customize the app's interface. Users felt that this would allow them to easily access and curate content that they want and when they want,

"*I think it will be more easier [sic] if people can find what they want directly, Malaysian people's, they're very picky they just want it, if they want to know the thing, they will just want to know that thing they don't like to go to everything so find the info.*"

[MSM, FG 3]

Some stakeholders suggested for an inclusion of a section where MSM could keep a record of their sexual activity, leading to more targeted counseling and treatment recommendations.

*"Yeah, something like my sex diary, so where they [MSM] could actually record the date, and then the type of, you know—the oral, anal, rimming, or anything like that, you could just click on it, and you know. It's like a fun thing to do, but then, at the same time, it is very, very vital because it helps a lot with the consultation process."*

[Stakeholders, FG 1]

## Caveats and concerns

### Culturally relevant

An important consideration that participants reported was that care needed to be taken to make sure every feature was tailored to be culturally relevant and tailored to a Malaysian socio-cultural context. Importantly participants reported that the app development must consider the lived experiences and perspectives of the LGBTQ+ community and consider the stigma and other barriers related to delivering sexual health services to Malaysian MSM:

*"So, something like a way I can get HIV testing, where I can get PrEP from then also additionally like which clinics are LGBT friendly will be nice as well, especially in Malaysia. Then yeah probably if I think I didn't see this but I think if it's possible probably get the pricing on the testing and the prep medication in certain clinics would be great as well, and these people can prepare beforehand about how much they are about to spend there, So yes, I do think that it's a majority of the options on the current app is already good, but I think we can add it to fill or fit more Malaysian standards, I guess."*

*[MSM, FG 2]*

## Too many features may jeopardize user-friendliness

Stakeholders pointed out that while having a variety of functions is desirable, a balance must be struck between services offered and overall user-friendliness. One stakeholder commented that having too many features can make an app difficult to use and maintain:

**"***I mean I have been kind of toying with, you know, kind of. . . kind of I.T. stuff in the clinic for a while, and I think I guess, one of the things to be sensitive about is that it actually takes more maintenance than we think, so like. Like you can design a very, very complicated APP that does everything. And usually, when you try to do too many things, then it gets uglier and uglier because it's really hard to make things look pretty on one screen. So that's something to think about."*

*[Stakeholders, FG1]*

## Confidentiality/Privacy

MSM participants expressed many concerns over anonymity and whom would be able to access their personal health information (PHI). For example, when asked about receiving test results, a participant reported, "I believe, like all of the information should be private and confidential." They shared greater willingness to use the app if proper transparency was practiced and the purposes for various information requested was made clear. Moreover, patients stressed the importance that reminders appear in a way that protects their privacy. One stakeholder discussed the use of code words to replace app identifiers, while another suggested that one's browsing? history should be irretrievable. An MSM participant emphasized the importance of discreteness to others who may see the app but not exactly know the purpose of it:

*"I think if the target group are the LGBTQI community, I think we just come up with more discreet name, something I mean something that it's just really health related, it doesn't have to be you know, like you know, gays r us but like something like you know. Prima care or something that is something health-related; it's very discreet, you know, and so, even if somebody sees like oh it's just a health app you know, like it helps monitor a heart rate."*

*[MSM, FG 2]*

The issue of safety measures in place was also introduced, as stakeholders wanted to know what security protocols were in place, such as data transmission, encryption methods, and data storage. Along such lines, some participants requested the use of multi-factor authentication, including password input and identity verification.

## Visually appealing and discreet

MSM participants were concerned whether the aesthetic of the app would appear to disclose sexual orientation. For example, one participant shared that the logo should not connote sexual orientation and another participant pointed out that the logo must be discreet and that the app should be geared toward everyone in the MSM community–including those who are in situations where privacy and disclosure of their sexual orientation could be very personally costly to the user:

*I can say, and in terms of this discreteness, I'm not really sure what is the logo that you use for the apps, but you know from whom, is it?"*

[MSM, FG 3]

In addition, stakeholders discussed the need to make the app aesthetically pleasing. One physician discussed the need for visual appeal within the app to create more enticement to continue usage:

"*Very superficial but it kind of looked very bland and the one thing that I know from my clients, is that they prefer something a bit more interactive than that, something with a bit more pop. And yeah, that kind of looks very clinical, very drab and I don't think that's gonna attract MSM users here at least yeah.*"

*[Stakeholder, FG3]*

In sum, for participants concerns related to visual appeal and discretion go hand in hand. The app aesthetics and presentation must be deliberately discreet to avoid connotations with sexual orientation to make the app usable for members who may be in situations where safe disclosure is not possible.

### Language

Participants pointed out that the Malaysian cultural context is rich and diverse. The region has several dominant languages which must be reflected in the app. Stakeholders pointed out that language is often a barrier in terms of serving clients in this region and participants reported that language barriers were a significant issue. For example, in response to a group prompt to further elucidate issues related to language, participants were asked to rate on a scale of 1–10 how big of an issue language barriers are, and a stakeholders reported they were significant with a participant saying:

"*I would say six or seven—I think language is definitely a big issue.*"

[Stakeholder, FG 2]

Language barriers were a shared concern among MSM and stakeholder participants and members of both groups supported a platform that would be offered in multiple languages.

### Discussion

Our findings provide important insights into developing a smartphone app as a tool to promote HIV testing and PrEP uptake among Malaysian MSM. Key categories included preferences for the app's functions (e.g., informational multimedia resources), engagement strategies (e.g., interactivity, reminders), and concerns related to the app (e.g., cultural relevance, privacy). MSM and stakeholders expressed support and positivity about the potential for the app to engage a broader range of MSM in HIV prevention.

Participants voiced interest in a feature to document their behavior to inform a personalized HIV prevention plan (e.g., HIV testing, PrEP initiation) and notify users when they are due for follow-up care. Research from the Malaysian and broader Southeast Asian context shows that the low perceived risk of HIV is one of the key reasons MSM do not engage in regular HIV testing [5,29]. Therefore, a built-in app feature that helps users assess their HIV-associated risks could motivate them to get tested for HIV and initiate PrEP services. Work

regarding the acceptability of mHealth interventions among Malaysian MSM found that 81.4% of participants were willing to use an app to track their sexual activity [22]. Future research should examine the actual reported usage of activity tracking features to assess the actual adoption of the feature.

Participants expressed ambivalent feelings regarding the possibility of ordering health products, such as HIV self-test kits, through the app. Both MSM and stakeholders felt concerned about the lack of medical supervision and subsequent access to treatment but felt more at ease when presented with the option of pairing the kit with a video tutorial or live video consultations with a medical professional. This insight is supported by a study that found that HIV self-testing services with the highest acceptability among Malaysian MSM included phone or texting hotlines but that there was a need to ensure that testing services inevitably led to linkage to either prevention or treatment follow-up services [30]. Taken together, these findings suggest that to increase the uptake of HIVST, MSM should be provided with the option to communicate with a professional about their result.

Both MSM and stakeholders indicated the importance of incorporating telehealth consultations in the app as this feature could overcome access barriers related to the lack of clinics in rural areas by eliminating the cost and time of travel to better-serviced urban areas. Stakeholders, however, were concerned about the legality of providing certain services via telehealth and the challenge of verifying the identity of a telehealth patient. During the COVID-19 pandemic, concerns about medicolegal issues and consent were also cited as a top barrier to the adoption of telehealth among Malaysian physicians [31].

A challenge for the development of the app is the manner of communication between participants and physicians. MSM valued having several options for interaction with medical professionals through the app, including telehealth visits, a 24/7 live chat, and a forum to ask medical questions. These types of communication features are similar in nature to "patient portals" and other features that provide direct messaging between providers and patients and may be subject to similar issues. To work effectively and safely, these features should include explicit instructions on their use for patients as direct messaging features often fail to have an educational component that establishes guidelines and expectations for patients [32]. In addition, stakeholders expressed concerns that some of these features could encumber physicians with a heavy workload and introduce liability issues for the care of distressed and/or suicidal users seeking emergency support which is similar to concerns shared by doctors regarding other direct messaging features outside of scheduled appointments [33]. Future research should explore medicolegal issues surrounding telehealth and pilot different communication features to balance accessibility for MSM and acceptability for physicians.

Confidentiality and cost of services were important concerns for MSM. These findings resonate with findings that confidentiality and cost are common barriers to PrEP uptake among MSM in Malaysia [34] and further demonstrates that these concerns may deter use of other prevention services related to HIV and sexual health offered through an app. In terms of confidentiality, participants emphasized that the app icon and any on-screen notifications must be discreet and provide no indication of the user's sexual orientation. In addition, participants advised that any purchases made through the App (e.g., HIVST kits, safe sex supplies) should have the option of shipping to P.O. boxes to protect anonymity. Furthermore, MSM were concerned about the cost of any services and supplies provided by the App, consistent with previous studies that found that Malaysian MSM are only willing to use PrEP and HIVST if provided for free or at low cost [30,35]. In addition, some personalization features that we suggested by participants such as sexual activity diaries or other features that require sensitive data which create concerns around privacy and confidentiality. These sorts of features must be constructed in a way that is attentive to privacy concerns and does not require prompt details

(e.g., gender of partners, number of partners) that might jeopardize the safety of participants. While greater personalization and other features might benefit participants, it also is important to weigh the benefits of those features against the need for data safety, privacy, and security.

The current study is not without limitations. While participants were recruited through advertisements at community organizations, participants were also recruited through social media. Recruitment of MSM through social media platforms may have led to a study sample that is, on average, more technologically literate than the general MSM population and, thus, may lead to the overestimation of the acceptability of app features. Moreover, recruiting participants from social media may have led to a sample that was more open to interventions delivered via technology. Importantly, this strategy may underrepresent MSM community members who have more concerns with using GNS platforms and social media which may have led to a sample that is more open and positive about using technology for their sexual health needs. In addition, social desirability bias is common in FG sessions. Specifically, participants may feel pressure to conform to sentiments expressed by the group, leading participants to express agreement or disagreement that may not be consistent with their own beliefs to be well-received by other participants. This may have influenced to minimize social desirability concerns, researchers encouraged participants to use pseudonyms and keep their cameras off throughout the discussion. Finally, it should be noted that willingness to use a given feature may be disparate from actual use, and real-world usage must be assessed in future work.

## Conclusion

To our knowledge, this is the first study to qualitatively examine MSM and stakeholders' preferences for an integrated HIV prevention app. Our findings provide evidence that a digital intervention would not only be innovative but also welcomed by stakeholders and MSM alike. Findings from the current work resonate with studies related to telemedicine (e.g., medicolegal issues, physician burden) and sexual healthcare (e.g., confidentiality, cost) more broadly for MSM in Malaysia and build upon by understanding perspectives and preferences towards an app for HIV prevention services. Several concerns about the app that require further exploration include confidentiality, cost, communication between MSM and medical providers, and legal issues related to the provision of virtual medical services. Future research should pilot aspects of the interventions explored in this paper to determine real-world feasibility, acceptability, and uptake. For example, future studies could investigate whether certain suggested features (e.g., medication tracker, informational multimedia resources) are utilized in an actual app. In sum, this paper generates knowledge on potential features to be included and tested in the design of an app geared toward HIV prevention for MSM in Malaysia.

## Author Contributions

**Conceptualization:** Roman Shrestha.

**Formal analysis:** Lindsay Palmer, Francesca Maviglia, Beverly-Danielle Bruno.

**Funding acquisition:** Roman Shrestha.

**Methodology:** Roman Shrestha.

**Project administration:** Roman Shrestha.

**Supervision:** Roman Shrestha.

**Validation:** Roman Shrestha.

**Writing – original draft:** Lindsay Palmer.

**Writing – review & editing:** Jeffrey A. Wickersham, Kamal Gautam, Francesca Maviglia, Beverly-Danielle Bruno, Iskandar Azwa, Antoine Khati, Frederick L. Altice, Kiran Paudel, Sherry Pagoto, Roman Shrestha.

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
