## [Decision Letter · Decision Letter 0]

11 Jul 2024

PDIG-D-24-00220

User Preferences for an mHealth App to Support HIV Testing and Pre-exposure Prophylaxis Uptake among Men Who Have Sex with Men in Malaysia

PLOS Digital Health

Dear Dr. Shrestha,

Thank you for submitting your manuscript to PLOS Digital Health. After careful consideration, we feel that it has merit but does not fully meet PLOS Digital Health's publication criteria as it currently stands. Therefore, we invite you to submit a revised version of the manuscript that addresses the points raised during the review process.

Please submit your revised manuscript within 60 days Sep 09 2024 11:59PM. If you will need more time than this to complete your revisions, please reply to this message or contact the journal office at digitalhealth@plos.org. Please include the following items when submitting your revised manuscript:

We look forward to receiving your revised manuscript.

Kind regards,

Haleh Ayatollahi

Section Editor

PLOS Digital Health

Journal Requirements:

Additional Editor Comments (if provided):

Reviewers' comments:

Reviewer's Responses to Questions

**Comments to the Author**

1. Does this manuscript meet PLOS Digital Health’s publication criteria? Is the manuscript technically sound, and do the data support the conclusions? The manuscript must describe methodologically and ethically rigorous research with conclusions that are appropriately drawn based on the data presented.

Reviewer #1: Yes

Reviewer #2: Yes

2. Has the statistical analysis been performed appropriately and rigorously?

Reviewer #1: Yes

Reviewer #2: Yes

3. Have the authors made all data underlying the findings in their manuscript fully available (please refer to the Data Availability Statement at the start of the manuscript PDF file)?

Reviewer #1: Yes

Reviewer #2: Yes

4. Is the manuscript presented in an intelligible fashion and written in standard English?

Reviewer #1: Yes

Reviewer #2: Yes

5. Review Comments to the Author

Reviewer #1: The manuscript addresses an important and timely topic of developing a mHealth app for HIV prevention among MSM (Men Who Have Sex with Men) in Malaysia. 

The well-structured study provides valuable insights into user preferences, which are crucial for designing effective digital health interventions. 

However, further refinement could benefit several areas to enhance clarity, depth, and overall impact.

• Introduction: please include a more detailed discussion on existing mHealth interventions globally and their impact, setting the context for why this study is necessary.

• The abstract is comprehensive, but it is somewhat lengthy. Consider condensing it to ensure it succinctly captures the key points without overwhelming the reader.

• Methods: The recruitment process is well-explained, but more details on how participants were selected and any potential biases this might introduce would be beneficial. For instance, did the recruitment through social media platforms like Grindr and Hornet influence the sample's demographic characteristics?

• In the data analysis section, please explain the coding process and how themes were identified and validated.

• Results: The results section is thorough, presenting detailed preferences and concerns of MSM and stakeholders. However, it would be helpful to include more direct quotes from participants to vividly illustrate the themes.

• Table 1 and Table 2 provide helpful information but could be more effectively integrated into the text. Discussing these tables in more detail within the narrative would enhance their relevance.

• Discussion: Please strengthen this section by linking the results more explicitly to the existing literature on mHealth interventions for HIV prevention. This would help to situate the findings within the broader field.

• Please discuss on the potential impact of these limitations in more detail. For example, how might the recruitment method have biased the findings?

• Please expand the section on the conclusion to provide more specific recommendations for future research and practical implementation of the app.

• It would be valuable if you could elaborate more on how ethical concerns, especially regarding confidentiality and data security, were addressed during the study.

• The manuscript presents significant findings that can contribute to the development of an effective mHealth app for HIV prevention among MSM in Malaysia.

Reviewer #2: 1. The manuscript meet PLOS Digital Health’s publication criteria and is technically sound, and the data support the conclusions. The manuscript describes methodologically and ethically rigorous research with conclusions that are appropriately drawn based on the data presented. 

2. The statistical analysis has been performed appropriately and rigorously

3.Authors made all data underlying the findings in their manuscript fully available

4. The manuscript is presented in an intelligible fashion and written in standard English with minimal grammatical errors that need to be corrected by the author.

5. In the Methods section, you mentioned that there were 36 participants in the study, specifically 20 MSM and 16 community stakeholders. However, you also mention clinical stakeholders later on. Were the clinical stakeholders a subset of the 16 community stakeholders? Please clarify this point.

6. What did you mean by this statement "To be eligible,community and clinical stakeholders must have been involved with providing HIV related services to MSM"? Please improve the write up under participants and settings section.

7. Based on your methodology, it appears that this was a mixed methods study design. However, this is not clearly articulated in your write-up. Please review this section to clearly include and describe the study design used for this study.

8. I noticed that you referred to the COVID-10 pandemic in the discussion section, which is incorrect. Please review and update it to reflect the correct year.

6. PLOS authors have the option to publish the peer review history of their article (what does this mean?). If published, this will include your full peer review and any attached files.

**Do you want your identity to be public for this peer review?** For information about this choice, including consent withdrawal, please see our Privacy Policy.

Reviewer #1: Yes: Dr K Madan Gopal

Reviewer #2: Yes: Kinene Andrew

---

## [Decision Letter · Decision Letter 1]

16 Sep 2024

User Preferences for an mHealth App to Support HIV Testing and Pre-exposure Prophylaxis Uptake among Men Who Have Sex with Men in Malaysia

PDIG-D-24-00220R1

Dear Dr. Shrestha,

We are pleased to inform you that your manuscript 'User Preferences for an mHealth App to Support HIV Testing and Pre-exposure Prophylaxis Uptake among Men Who Have Sex with Men in Malaysia' has been provisionally accepted for publication in PLOS Digital Health.

Best regards,

Haleh Ayatollahi

Section Editor

PLOS Digital Health

Reviewer Comments (if any, and for reference):

Reviewer's Responses to Questions

**Comments to the Author**

1. If the authors have adequately addressed your comments raised in a previous round of review and you feel that this manuscript is now acceptable for publication, you may indicate that here to bypass the “Comments to the Author” section, enter your conflict of interest statement in the “Confidential to Editor” section, and submit your "Accept" recommendation.

Reviewer #1: All comments have been addressed

Reviewer #2: All comments have been addressed

2. Does this manuscript meet PLOS Digital Health’s publication criteria? Is the manuscript technically sound, and do the data support the conclusions? The manuscript must describe methodologically and ethically rigorous research with conclusions that are appropriately drawn based on the data presented.

Reviewer #1: Yes

Reviewer #2: Yes

3. Has the statistical analysis been performed appropriately and rigorously?

Reviewer #1: Yes

Reviewer #2: Yes

4. Have the authors made all data underlying the findings in their manuscript fully available (please refer to the Data Availability Statement at the start of the manuscript PDF file)?

Reviewer #1: Yes

Reviewer #2: Yes

5. Is the manuscript presented in an intelligible fashion and written in standard English?

Reviewer #1: Yes

Reviewer #2: Yes

6. Review Comments to the Author

Reviewer #1: (No Response)

Reviewer #2: The authors have successfully addressed all my previous comments, and I am confident that this study will make a valuable contribution to the body of knowledge, particularly in research related to MSM (Men who have Sex with Men) groups. I commend the authors for reaching this important milestone in their research journey.

I also appreciate the authors' efforts in refining the manuscript, including correcting grammatical issues. Furthermore, the statistical analysis has been conducted with rigor and precision, adding robustness to the findings.

I look forward to seeing this work published.

7. PLOS authors have the option to publish the peer review history of their article (what does this mean?). If published, this will include your full peer review and any attached files.

**Do you want your identity to be public for this peer review?** For information about this choice, including consent withdrawal, please see our Privacy Policy.

Reviewer #1: No

Reviewer #2: **Yes: **Kinene Andrew
